# Follow-up of Interleukin 6 and Other Blood Markers during the Hospitalization of COVID-19 Patients: A Single-Center Study

**Maritza P. Garrido** [1,2,*], **Varsha Vaswani** [3], **Katherinne Contreras** [3], **Marcela Barberán** [4], **Manuel Valenzuela-Valderrama** [5], **Diana Klajn** [6], **Carmen Romero** [1,2], **María Jesús Vial Covarrubias** [7] and **Rodrigo Alfredo Cornejo** [8]

1  Laboratorio de Endocrinología y Biología de la Reproducción, Hospital Clínico Universidad de Chile, Santiago 8380456, Chile
2  Departamento de Obstetricia y Ginecología, Facultad de Medicina, Universidad de Chile, Santiago 8380453, Chile
3  Sección de Endocrinología y Diabetes, Hospital Clínico Universidad de Chile, Santiago 8380456, Chile
4  Clinica las Condes, Sección Endocrinología, Santiago 7591047, Chile
5  Laboratorio de Microbiología Celular, Instituto de Investigación y Postgrado, Universidad Central de Chile, Santiago 8320000, Chile
6  Hospital General de Agudos Enrique Tornú, Comité de Docencia e Investigación, Universidad de Buenos Aires, Buenos Aires C1427ARN, Argentina
7  Servicio de Laboratorio Clínico, Hospital Clínico Universidad de Chile, Santiago 8380456, Chile
8  Unidad de Pacientes Críticos (UPC), Departamento de Medicina, Hospital Clínico Universidad de Chile, Santiago 8380456, Chile
*  Correspondence: mgarrido@hcuch.cl

**Abstract:** COVID-19 is a recent respiratory illness with high morbidity and mortality; therefore, the study and characterization of blood markers associated with the improvement or deterioration of COVID-19 patients are crucial. This study compared levels of interleukin 6 (IL-6), procalcitonin (PCT), D-dimer, cortisol, dehydroepiandrosterone sulfate (DHEA-S), c-reactive protein (CRP), 25-OH vitamin D, anti-SARS-CoV-2 IgG antibodies, and viremia in mild–moderate and severe–critical COVID-19 patients. In addition, the time course of blood markers was studied in severe–critical cases. The results show that levels of IL-6, PCT, D-dimer, and CRP, the cortisol/DHEA-S ratio, as well as positive viremia and anti-Spike IgGs were higher in severe–critical patients requiring hospitalization. During follow-up, most severe–critical cases displayed similar time patterns of IL-6 and viral load, whereas anti-SARS-CoV-2 antibody curves showed an inverse pattern. A decrease in IL-6 levels was associated with the improvement of COVID-19 patients, mostly through a reduced oxygen requirement. This preliminary study suggests that an increase in serum IL-6, PCT, D-dimer and CRP levels and the cortisol/DHEA-S ratio could support the selection of patients with poorer prognosis and the need for an intensive or alternative treatment. Additionally, changes in IL-6 during hospitalization were associated with changes in patient's status mainly with a decrease in oxygen requirements, which indicates that serial measurements of IL-6 could predict the outcome of severe–critical patients with COVID-19 pneumonia.

**Keywords:** SARS-CoV-2; COVID-19; Interleukin 6; cortisol; DHEA-S; blood markers

## 1. Introduction

Coronavirus disease 2019 (COVID-19), caused by severe acute respiratory syndrome coronavirus 2 (SARS-CoV-2), is a disorder with a variable course. Approximately 20% of cases develop severe illness and require hospitalization with oxygen therapy or mechanical ventilation [1–3]. As a recent disease, its physiopathology is not fully known. Several blood markers have been proposed to support patient management and health care professionals' decisions, including interleukin 6 (IL-6) [4–6], procalcitonin (PCT) [7–9], D-dimer [10–12],

c-reactive protein (CRP) [13–15], and viremia [16–18]. In addition, because of its antimi-crobial, anti-inflammatory, and immunomodulatory properties, it is discussed whether hypovitaminosis D could be associated with severe disease, increased intensive care unit (ICU) admission and the mortality rate of COVID-19 patients [19,20].

In the endocrine system, evidence suggests that the adrenal gland is among the most impaired organs in severe–critical COVID-19 patients. That is because (1) viremia could be assessed in these patients [21], by which SARS-CoV-2 could spread to other tissues, including endocrine glands; (2) some amino acid sequences of SARS virus have molecular mimics of the host adrenocorticotropic hormone (ACTH) which could lead to the produc-tion of anti-ACTH antibodies [22], disturbing adrenal function; and (3) the corticoid therapy widely used in these patients could impair adrenal function after hospital discharge [23]. Therefore, the study of adrenal status during SARS-CoV-2 infection is relevant.

Most studies have proposed that measurements of these blood markers could predict patient outcomes and be useful during hospitalization care. However, there are no studies of these markers in the Chilean population nor of their kinetics during hospitalization of COVID-19 patients, which makes it difficult for routine use of such markers. In this study, we reported levels of several blood markers in mild–moderate and severe–critical patients with COVID-19. In addition, time patterns of some endocrine and inflammatory markers along with the course of viremia and anti-SARS-CoV-2-neutralizing antibodies were assessed in patients with the severe–critical disease (COVID-19 pneumonia).

## 2. Materials and Methods

### 2.1. Patients and Samples

Blood samples of 31 patients with respiratory failure, 18 years or older, were collected from August 2020 to February 2021. Patients included in the study were admitted to the emergency room of the Hospital Clínico Universidad de Chile and had a positive PCR test for SARS-CoV-2 on a nasopharyngeal swab. Fourteen patients developed a severe or critical disease (COVID-19 pneumonia) and were admitted to hospital care from the emergency room. The remaining 17 mild–moderate patients were discharged in less than 24 h from the emergency room after stabilization (Figure 1). None of the patients had been vaccinated at the time of this study.

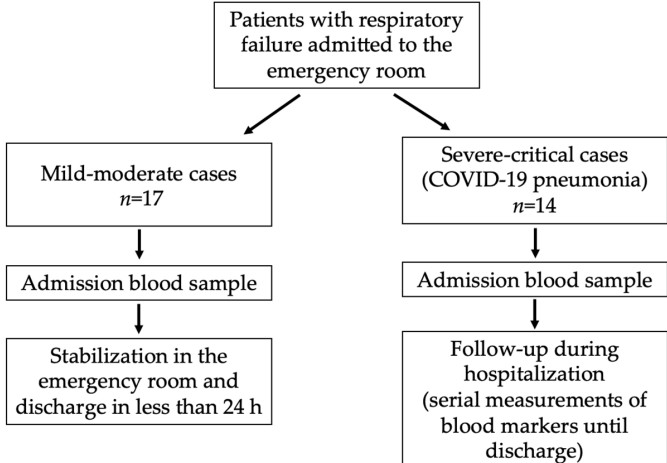

**Figure 1.** Patient recruitment flowchart.

Patient conditions that could strongly modify the blood markers studied, including cancer, and/or myocardial, hepatic, or renal insufficiency were excluded from this study. Patients with an asymptomatic COVID-19 infection that were admitted for non-related diseases were also excluded.

Serum samples were used to perform all the measurements. Venous blood sampling was initially performed for patient clinical monitoring. Blood specimens were recollected in

yellow tubes of 3 mL (without anticoagulant) with a clot activator and gel to obtain serum samples. Collection tubes were centrifuged at 2500 rpm for 10 min and serum aliquots were stored at $-80\,°C$ until their use. In severe–critical cases, the follow-up samples were collected every 2 or 3 days until hospital discharge. After reviewing each patient's clinical record, samples temporarily related to positive or negative clinical events were processed.

A chest computed tomography was performed in severe–critical COVID-19 patients during their hospital admission. Images were analyzed using the software U-Net (U-Net: Convolutional Networks for Biomedical Image) [24] to obtain the quantification of ground-glass opacity of infection regions.

All clinical parameters were obtained by clinical record review. Supplementary Table S1 summarizes these parameters.

### 2.2. Negative and Positive Events

Severe–critical patients were monitored for positive or negative events. The following were considered positive events: transfer to a less complex therapeutic unit (from the ICU to a general ward), a substantial decrease in oxygen requirements (change from mechanical ventilation to an oxygen mask, switch from high-flow to low-flow system), interruption of oxygen delivery, or hospital discharge.

On the other hand, these were considered negative events: transfer to a more complex therapeutic unit (from a general ward to the ICU) and a substantial increase in oxygen requirements (need for mechanical ventilation or switch from low-flow to high-flow system).

### 2.3. Blood Measurements

Serum interleukin 6 (IL-6), procalcitonin (PCT), and D-dimer were measured by chemiluminescence using Maglumi 800 equipment (Snibe Co, Shenzhen, China). Levels of cortisol, 25-OH vitamin D, dehydroepiandrosterone sulfate (DHEAS), and anti-SARS-CoV-2 IgG antibodies were measured by electrochemiluminescence using Cobas e601 equipment (Roche Diagnostics, Basel, Switzerland). CRP levels in each patient were obtained from clinical records because it was part of the continuous monitoring of patients. CRP measures were performed in VITROS 5600 platform (Ortho Clinical Diagnostics, Raritan, NJ, USA) and values > 3.0 mg/L indicate a high risk of inflammation or infection. All measurements were performed under the manufacturer's specifications.

An aliquot of 500 µL of serum was used to perform automatized assays, which had an inter-assay variation coefficient below 6%. Table 1 summarizes the analytical characteristics of blood tests.

**Table 1.** Analytical characteristics of tests used to measure blood markers. IL-6: interleukin 6. PCT: procalcitonin. DHEA-S: dehydroepiandrosterone sulfate. FEU: fibrinogen equivalent units. II, III: kits of second and third generation, respectively. S: specific IgG anti-Spike protein. y: years. F: female. M: male. * Reference values for matinal cortisol. ** = reference values of 5–95 percentile according to assay datasheet.

| Blood Markers Kits | Measuring Range of Test | Reference Values (Healthy Population) | Units |
|---|---|---|---|
| Maglumi IL-6 | 0.5–5000 | <7.0 | pg/mL |
| Maglumi PCT | 0.01–100 | <0.05 | ng/mL |
| Maglumi D-dimer | 100–10,000 | <500 | ng FEU/mL |
| Elecsys Anti-SARS-CoV-2 S | 0.40–250 | <0.80 | U/mL |
| Elecsys Cortisol II | 0.05–63.4 | 5.0–25 * | ug/dL |
| Elecsys DHEA-S | 0.1–1000 | 20–44 y: 60.9–340 (F)/88.9–492 (M) ** <br> 45–74 y: 35.4–256 (F)/33.6–331 (M) ** <br> >75 y: 12.0–154 (F)/16.2–123 (M) ** | ug/dL |
| Elecsys 25-OH Vitamin D III | 3.0–120 | >20 (sufficient) <br> >30 (optimum) | ng/mL |

### 2.4. Detection of Viremia

A volume of 150 μL of serum was diluted 1:1 with RPMI 1640 medium (Gibco, Thermo Fisher Scientific, Waltham, MA, USA). Viral RNA was obtained by organic extraction using the TrizolTM-based method and then precipitated from the aqueous phase using isopropanol 50% and Glycoblue as co-precipitant agent (Thermo Fisher Scientific, Waltham, MA, USA), as previously described [18,21]. RNA was resuspended in 20 μL of nuclease-free water and quantified. Detection of SARS-CoV-2 RNA was performed by RT-qPCR using the Takyon™ One-Step Kit Converter plus the 2X Takyon™ qPCR dTTP MasterMix (Eurogentec, Seraing, Belgium), according to the manufacturer's instructions. N1- and N2-specific regions of the viral nucleoprotein and the human ribonuclease (RNase) P genes were amplified using the primers and probe sets included in the 2019-nCov RUO qPCR probe kit (Integrated DNA Technologies, Coralville, IA, USA). In addition, the nCoV nucleocapsid gene present in the 2019-nCoV_N Positive Control (Integrated DNA Technologies) was used as the control. Changes in viral load were evaluated by the $2^{-\Delta\Delta CT}$ method [25], considering the initial load as 100%.

### 2.5. Ethics Issues

The use of blood samples, clinical records, and data management were approved by the Hospital Clinico Universidad de Chile Ethics Committee (Record N° 49, 2020).

### 2.6. Statistical Analysis

Data were expressed as the means with standard deviation (SD) or the medians with interquartile range (IQR). Student's t-test or the Mann–Whitney test was used to compare biomarkers in severe–critical patients and mild–moderate patients at admission. To compare biomarker samples of the same patient at different times during the course of the disease, a Wilcoxon test was performed. To assess the association between biomarker values and clinical/radiological scores a Spearman correlation test was applied. A $p$-value < 0.05 was considered statistically significant.

## 3. Results and Discussion

### 3.1. Levels of Blood Markers in COVID-19 Patients Requiring and Not Requiring Hospitalization

No significant differences in the demographic characteristics were observed between the groups studied (Supplementary Table S1). Serum levels of IL-6 were higher in severe–critical patients that developed COVID-19 pneumonia who were admitted to hospital care (severe–critical cases), compared to mild–moderate cases not requiring hospitalization. Similar results were observed in PCT, D-dimer, and CRP measurements (Figure 1, Supplementary Table S2). In addition, IL-6 levels showed a positive correlation with the radiological finding of ground-glass opacity (GGO) in severe–critical COVID-19 patients (r = 0.65). GGO values of each patient were detailed in Supplementary Table S3. Because GGO is a quantification of hazy gray areas observed in computed tomography images, our results suggest that IL-6 levels could be related to pulmonary involvement in COVID-19 patients.

Severe–critical patients had lower DHEA-S blood levels compared to mild–moderate cases, as well as higher cortisol/DHEA-S ratios (Figure 2, Supplementary Table S4). It is important to highlight that these measurements were performed in the emergency room, hours before their hospitalization when none of the patients had received corticoid therapy and that both groups of patients (mild–moderate and severe–critical) had median cortisol levels < 20 μg/dL. This cut-off is used as a suspicion of adrenal insufficiency and argues for the use of corticosteroids in patients with community-acquired pneumonia [26]. Since it is reported that SARS-CoV-1 expresses amino acid sequences that are molecular mimicry of the host's ACTH [22], SARS-CoV-2 could affect the hypothalamic-pituitary-adrenal axis, similar to that observed with the original SARS virus and it could blunt the stress-induced cortisol rise, for instance, producing antibodies against the viral particles that would inadvertently decrease the circulating ACTH level. In this regard, autopsy studies performed on patients who died from SARS-CoV-1 showed degeneration and necrosis of

adrenal cortical cells, suggesting a direct cytopathic effect of the virus [27], and additionally, ACE receptors have been reported in the adrenal cortical cells [28].

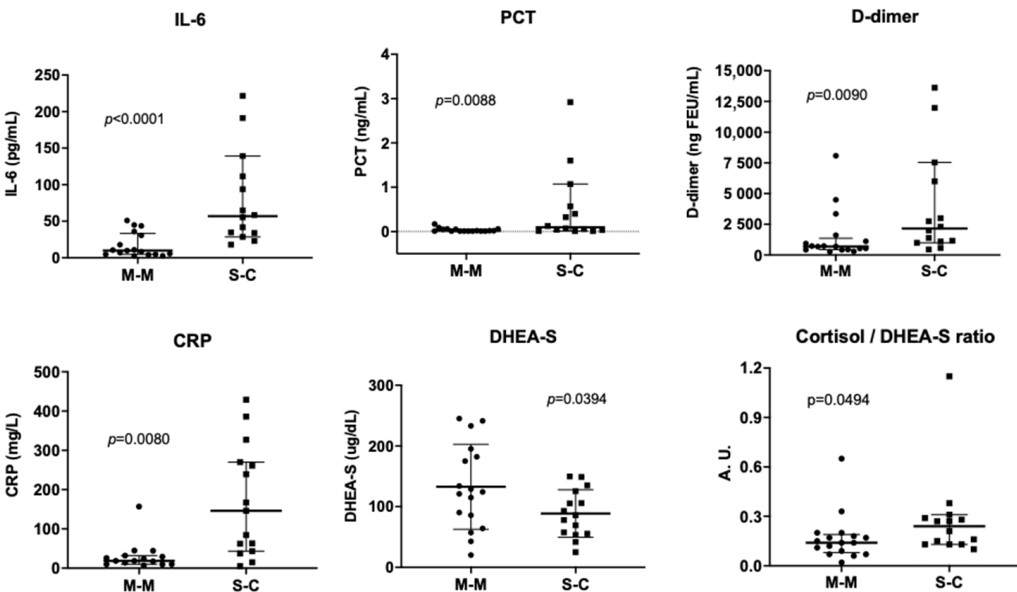

**Figure 2.** Levels of serum markers in patients with COVID-19. Panel A: levels of interleukin 6 (IL-6), procalcitonin (PCT), C-reactive protein (CRP), D-dimer, and dehydroepiandrosterone sulfate (DHEA-S) and the cortisol/DHEA-S ratio in patients with mild–moderate COVID-19 who were not hospitalized (M-M) and patients who developed the severe–critical disease (COVID-19 pneumonia) and were admitted for hospital care (S-C). Results are expressed as the mean and standard deviation (SD) for DHEA-S or the median with interquartile range (IQR) for all other markers. (Statistical analysis: t-test or the Mann–Whitney test according to data distribution).

According to the results, 24% of mild–moderate cases had positive anti-SASR-CoV-2 IgGs antibodies, compared to 50% among severe–critical patients. Viremia was detected in 31% of mild–moderate cases and in 93% of severe–critical patients. Among severe–critical patients admitted to an ICU, 77.8% had positive viremia at the time of admission. In addition, mean values of vitamin D in ICU and non-ICU patients were 17.0 and 22.6 ng/mL, respectively, in the range of deficiency and insufficiency (Supplementary Table S5). This is concordant with other studies which have found an association between vitamin D deficiency and severity/mortality of COVID-19 [19,29–31] and that low levels of vitamin D have been inversely correlated with high IL-6 [19]. In this context, a report showed that patients with COVID-19 and hypovitaminosis D that received a high dose of vitamin D (supplementation of 60,000 IU) significantly reduced some inflammatory markers such as CRP and IL-6 without any side effects [32]. Therefore, the supplementation with vitamin D in COVID-19 patients should be considered as a preventive strategy, particularly it should be a good clinical practice for high-risk patients (elderly with diabetes and obesity, and other comorbidities) [33] with hypovitaminosis D or vitamin D deficiency/insufficiency.

In a subanalysis of these patients, whether they were or were not admitted to an ICU, it appeared that IL-6, PCT, D-dimer, and CRP were higher in ICU patients (Supplementary Table S5). Additionally, it was found a positive correlation between cortisol levels at hospital admission and length of hospital stay in severe/critical patients (Supplementary Table S6).

*3.2. Time Patterns of IL-6 and Other Blood Markers during COVID-19 Course in Severe–Critical Patients*

A positive outcome during the follow-up of severe–critical patients was associated with a decrease in blood marker levels, particularly IL-6. Conversely, a high number of negative outcomes was related to an increase in IL-6, PCT, and D-dimer levels, as shown in Figure 3. Among the different blood markers studied, only the changes in levels of IL-6 had

a high correlation with the first (*p* = 0.017) and second (*p* = 0.023) events in the evolution of severe–critical COVID-19 patients (change in oxygen requirements) as shown in the Supplementary Table S7.

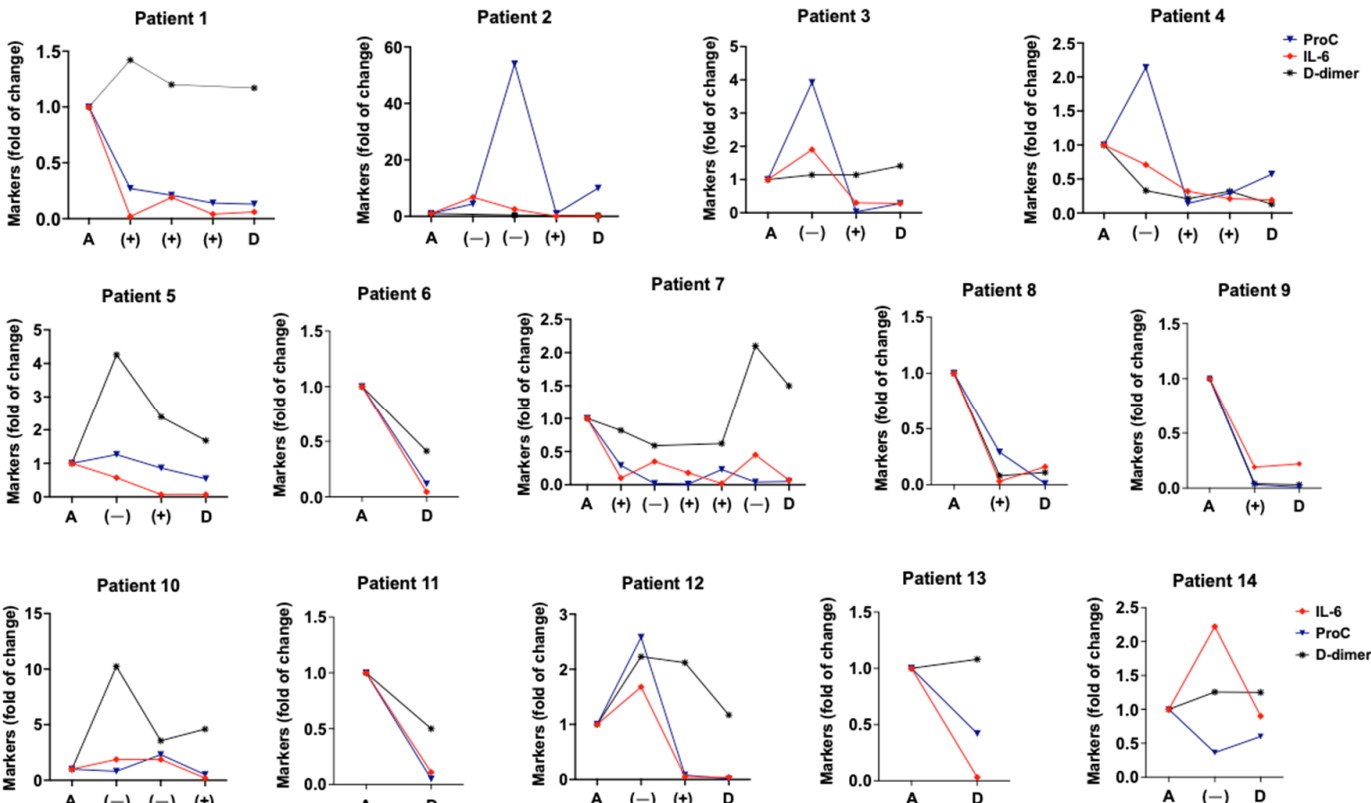

**Figure 3.** Time patterns of blood IL-6, PCT, and D-dimer of clinical course in severe–critical COVID-19 patients. Graphs show blood levels of interleukin 6 (IL-6), procalcitonin (PCT), and D-dimer during hospitalization of patients with COVID-19 pneumonia. Markers were measured at admission (A), discharge (D), and with any negative (−), and positive event (+), mainly related to the increase or decrease in oxygen therapy requirements (detailed in Section 2.2). Results are expressed as the fold change in the initial levels of each marker.

Additionally, 85% of patients showed neutralizing SARS-CoV-2 IgGs during the hospital stay. The time pattern of viral load and IL-6 was similar in most patients, while anti-SARS-CoV-2 IgGs showed an inverse pattern (Figure 4). Since IL-6 plays a key role in the viral immune response by activating and regulating the immune response during viral infections [34], our results suggest that the increase in IL-6 could be associated with the humoral response of COVID-19 patients, possibly promoting the generation of neutralizing antibodies against the virus. There is an ongoing debate about some possible complications of tocilizumab therapy (a monoclonal antibody that competitively inhibits the binding of IL-6 to its receptor) on the generation of humoral response in patients with COVID-19 pneumonia, given that IL-6 displays a pivotal role in the activation and regulation of the immune response during viral infections [34]. Our results indicate that a decrease in IL-6 levels is related to an exponential increase in anti-SARS-CoV-2 IgGs antibodies, suggesting that tocilizumab therapy would not interfere in the generation of these antibodies in COVID-19 patients, but rather, it could promote an early humoral response. This hypothesis is consistent with the results of a recent study that found that IL-6 blockade does not impair the viral-specific antibody responses [35] and patients with IL-6 above 30 ng/mL could benefit from tocilizumab administration [36].

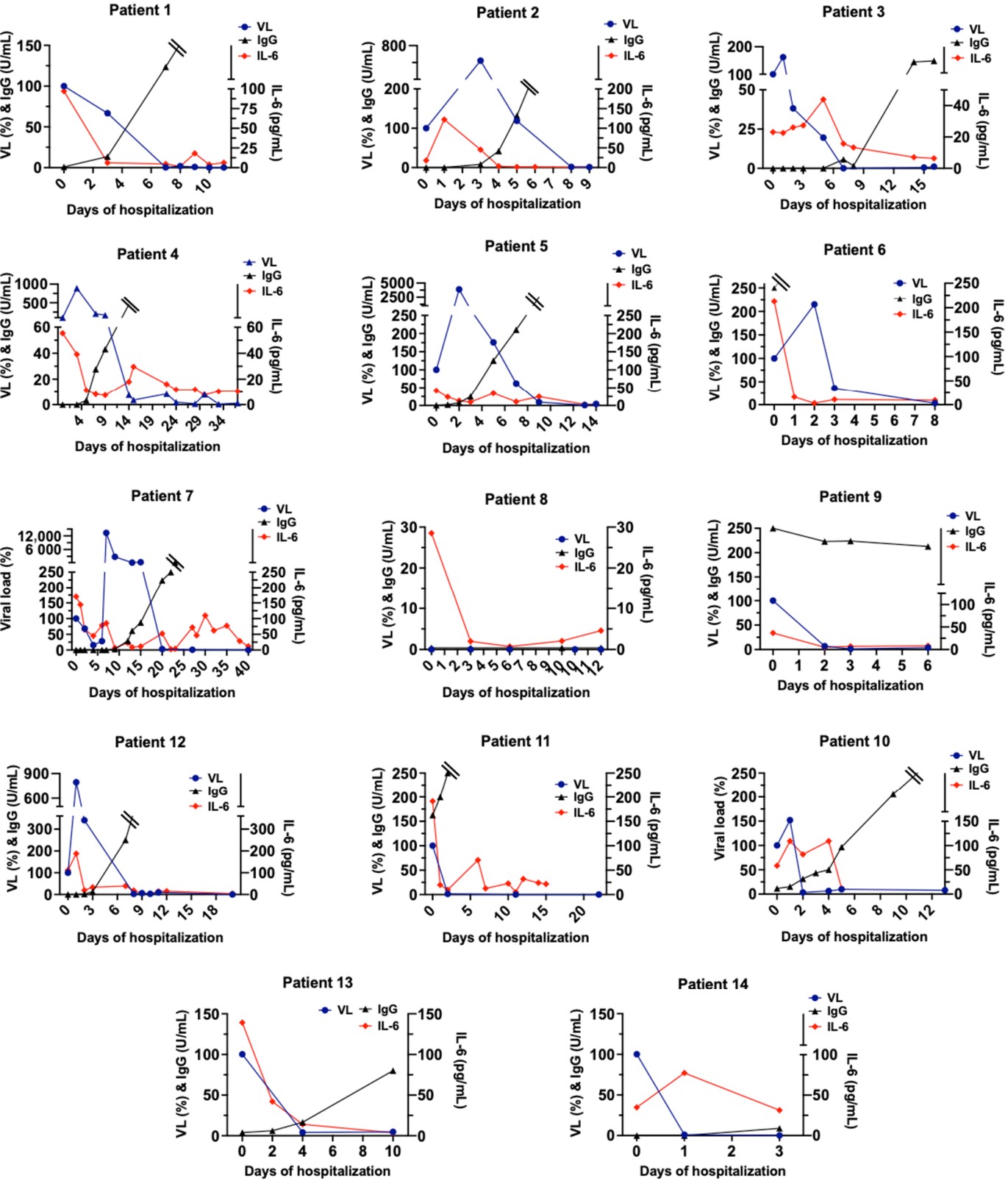

**Figure 4.** Time patterns of viremia, serum IL-6, and Anti-SARS-CoV-2-neutralizing antibodies (IgG) during severe/critical COVID-19 patient evolution. Graphs show blood levels of interleukin 6 (IL-6) Anti-SARS-CoV-2-neutralizing antibodies (IgG) and the virus load (viremia, VL) during hospitalization of patients with COVID-19 pneumonia. Markers were measured on different days until discharge. Double line: values of IgG over 250 U/mL (upper limit of the technique).

On the other hand, we found that the viral load of severe–critical patients with positive viremia decreased up to undetectable levels on day 9 [6–13 days] (Figure 4). International evidence has shown that extremely high levels of IL-6 were closely related to the frequency of viremia and vital signs of COVID-19 patients [37], and that viremia is associated with inflammatory, but not cardiovascular, biomarkers in hospitalized COVID-19 patients [38]. Therefore, viremia could be displaying an important role in IL-6 increase and thus in the humoral response of COVID-19 patients, accounting for the need for hospitalization.

The exponential increase in anti-SARS-CoV-2 IgG was observed earlier (mean 3.6 days) in severe–critical patients admitted to the ICU than in those not admitted (mean 8.8 days) (Supplementary Figure S1).

In severe–critical patients, IL-6, PCT, D-dimer, DHEAS, and cortisol levels decreased significantly at hospital discharge compared with the levels at admission (Supplementary Figure S2). These results showed an evident adrenal insufficiency during hospital discharge of several severe–critical patients (mean of 4.0 ug/dL, IQR [0.9–14.8]). Finally, IL-6 levels are highly correlated with CRP values (Supplementary Figure S3), a marker widely used for monitoring severe–critical patients [39].

## 4. Conclusions

Il-6, PCT, D-dimer, CRP, and DHEA-S/cortisol levels were higher in severe–critical patients requiring hospitalization. Excluding DHEA-S, all these biomarkers were higher in patients admitted to the ICU compared with those not admitted (hospitalized and non-hospitalized). Therefore, they could indicate a poorer prognosis and the need for an intensive or alternative treatment.

The higher cortisol/DHEA-S ratio observed in severe–critical patients before any corticotherapy would indicate adrenal disruption. In addition, the adrenal insufficiency observed during hospital discharge of severe–critical patients suggests that patients should be monitored by an endocrinologist after hospitalization.

The follow-up of patients hospitalized showed a decrease in IL-6 associated with an increase in viremia and a decrease in the humoral response. Changes in IL-6 during hospitalization were associated with changes in the patient's status, mainly with a decrease in oxygen requirements. This finding should be completed with new studies with a closer follow-up of IL-6 values in severe–critical patients. This will allow determining how long it takes the change in IL-6 to begin before the change in the patient's status.

**Supplementary Materials:** The following supporting information can be downloaded at: https://www.mdpi.com/article/10.3390/covid2110114/s1, Supplementary Table S1. Clinical characteristics of COVID-19 patients. Supplementary Table S2. Blood markers at hospital admission of mild–moderate and severe–critical patients (N = 31) and discharge of severe–critical patients. Supplementary Table S3. Ground-glass opacity (GGO) percent on chest computed tomography of 14 severe–critical COVID-19 patients. Supplementary Table S4. Cortisol/DHEA-S ratio in mild–moderate and severe–critical patients with COVID-19 on admission to the emergency room. Supplementary Table S5. Blood markers in ICU and non-ICU patients. Supplementary Table S6. Association between initial levels of blood markers on admission to the emergency room and hospital stay length. Supplementary Table S7. Association between the first positive or negative event and the levels of blood markers on the day of the event. Supplementary Figure S1. Days up to the exponential change in anti-SARS-CoV-2-neutralizing antibodies (IgG) during hospitalization of admitted to the ICU and not admitted to the ICU severe–critical patients with COVID-19. Supplementary Figure S2. Correlation between IL-6 and CRP levels in severe–critical patients with COVID-19. Supplementary Figure S3. Blood markers in hospital admission and discharge of severe–critical patients with COVID-19.

**Author Contributions:** Conceptualization, M.P.G. Sample obtention, data collection and data curation, M.P.G., V.V., K.C., C.R., M.J.V.C., M.V.-V. and R.A.C. Investigation, M.P.G., M.B. and R.A.C. Data analysis, M.P.G. and D.K. Writing—original draft, M.P.G. Writing—review and editing, M.P.G., M.B., V.V., M.V.-V., C.R. and D.K. All authors have read and agreed to the published version of the manuscript.

**Funding:** This research was funded by Universidad Central de Chile, grant number Project CIP2019015 (M.V.-V.), and the APC was funded by Hospital Clínico Universidad de Chile.

**Institutional Review Board Statement:** This study was conducted in accordance with the Declaration of Helsinki and approved by the Institutional Ethics Committee of Hospital Clínico Universidad de Chile (Record N° 49, 2020) for studies involving humans.

**Informed Consent Statement:** Since this was a retrospective study, the use of samples, clinical record review, and data management were approved by Institutional Ethics Committee.

**Data Availability Statement:** The datasets generated during the current study are available from the corresponding author upon reasonable request.

**Acknowledgments:** The authors would like to thank Snibe Diagnostic and Roche Chile LTDA. who donated reagents to determinate serum markers, and Jessica Arzola for her technical assistance.

**Conflicts of Interest:** The authors declare no conflict of interest.

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
