# Peer review of "Follow-up of Interleukin 6 and Other Blood Markers during the Hospitalization of COVID-19 Patients: A Single-Center Study"

_covid, doi:10.3390/covid2110114_

Round 1

Reviewer 1 Report

It is a very nice and important study, but I think that you need more than 31 patients in the study and then you can make the conclusion. Maybe it is not COVID disease the sample size can be acceptable, but you collected the patients 7 months and you have only 31 patients.

Author Response

Point-by-point response to reviewers’ comments 

Reviewer 1

English language and style are fine/minor spell check required

Answer: We would like to thank the reviewer for their comments. As recommended, we reviewed the manuscript to detect and correct language mistakes. Please, check the changes highlighted in yellow.

It is a very nice and important study, but I think that you need more than 31 patients in the study and then you can make the conclusion. Maybe it is not COVID disease the sample size can be acceptable, but you collected the patients 7 months and you have only 31 patients.

Answer: Although we agree with the reviewer's comment, it is important to highlight that we had samples from more patients. However, several of these patients had comorbidities that strongly affected the levels of the studied markers per se. Other patients did not have symptoms compatible with COVID-19 (they were hospitalized due to other pathologies and turned out to be infected with SARS-CoV-2). Since they did not meet the inclusion criteria, they were excluded from the study. Although our work was carried out with a small number of patients, the differences found among the markers were of great magnitude, so the changes turned out to be statistically significant despite this limitation. It is also for this reason that the results were presented as a brief report.

Reviewer 2 Report

Dear Autors,

This is an interesting study and the article is generally well written and structured.

However, in my opinion the document has some shortcomings with regard to some data

and text analysis. Below I have provided some observations and suggestions for the

presentation of the materials and methods and the results, which appear to be non-fluid.

In attached the minor revision

Best Regards

Author Response

Point-by-point response to reviewers’ comments

Reviewer 2

Moderate English changes required

Answer: We would like to thank the reviewer for the comments about our work. As recommended, we reviewed the manuscript to detect and correct inconsistencies in language and grammatical mistakes. Please, check the changes highlighted in yellow.

Material and Method were not clear: better specify the type of biological sample used (plasma or serum or whole blood, volume) and relative protocol of isolation, assay used for detection of marker; there is a range of value used for markers of interest; what are positive or negative clinical events considered.

Answer: We added and clarified this information in the methodology section.

There is a control group? For example patients without COVID infections or volunteers

Answer: Our study did not include a control group (without COVID-19). However, the reference values for non-pathologic population have been previously set. We added the reference values of each marker in the methodology section for a better understanding of our results.

The tables of Supplementary data were not clear. Below we suggest an example Characteristic Patients (n = xx) % p-value Age, years Median Range. All tables should have similar layout and all data of patients group should be indicated

Answer: We performed the suggested changes in the supplementary material.

The number of figures indicated in the text does not correspond to the legend

Answer: We checked and corrected the figure legends and cross-references in the manuscript.

Check the graphs fig 2 (line 142), the axes are different

Answer: The axes of Figure 2 were edited.

We could clarify the results shown in paragraph 3.2 (“Time-patterns of IL-6 and other blood markers during COVID-19 course in severe-critical patients”) and legend figure 3.

Answer: We reviewed paragraph 3.2 and some changes were incorporated in order to clarify these results.

Could be review the abbreviations

Answer: We checked and standardized the abbreviations along the text.

Have you thought about evaluating the expression of the Tissue Factor (TF), as a marker of oxidative damage?

Answer

This idea is very interesting since current knowledge suggests that overexpression of tissue factor (TF) in COVID-19 patients could be related to the pathogenesis of the disease.

The overexpression of TF was described in the lungs of COVID-19 patients (J. Thromb. Haemost. 2021 Sep;19(9):2268-2274). In addition, SARS-CoV-2 infection promotes a prothrombotic state and the excessive production of reactive oxygen species (ROS), which could exacerbate the host immunopathological response (EBioMedicine. 2022 Mar;77:103893; Sci Rep 2022 May; 12, 10484)

Importantly, a high neutrophil to lymphocyte ratio (key component that produces oxidative stress) is higher in critically ill patients with COVID-19 and predicts in-hospital mortality (Thromb. Res. 2020. 192, 3–8). Reactive oxygen species induce a procoagulant state in endothelial cells by inhibiting tissue factor pathway inhibitor (J. Thromb. Thrombolysis. 2015 Aug;40(2):186-92) and increased levels of ROS may induce TF gene expression (Circulation. 2002 Apr 30;105(17):2030-6). It is therefore highly probable that TF is increased in severe-critical patients and its measurement could indicate oxidative damage in these patients. Hence, the assessment of new possible disease markers such as TF would have to be measured in a future study with a design similar to our current research.